# Preclinical Study Using ABT263 to Increase Enzalutamide Sensitivity to Suppress Prostate Cancer Progression Via Targeting BCL2/ROS/USP26 Axis Through Altering ARv7 Protein Degradation

**DOI:** 10.3390/cancers12040831

**Published:** 2020-03-30

**Authors:** Hua Xu, Yin Sun, Chi-Ping Huang, Bosen You, Dingwei Ye, Chawnshang Chang

**Affiliations:** 1Department of Urology, Fudan University Shanghai Cancer Center, Shanghai 200032, China; 2Department of Oncology, Shanghai Medical College, Fudan University, Shanghai 200032, China; 3George Whipple Lab for Cancer Research, Departments of Pathology, Urology, and Radiation Oncology, University of Rochester Medical Center, Rochester, NY 14646, USA; 4The Wilmot Cancer Institute, University of Rochester Medical Center, Rochester, NY 14646, USA; 5Sex Hormone Research Center and Department of Urology, China Medical University, Taichung 404, Taiwan

**Keywords:** ABT263, androgen receptor, ARv7, Enzalutamide, prostate cancer

## Abstract

Background: The recently developed antiandrogen, Enzalutamide (Enz), has reformed the standard of care for castration resistant prostate cancer (CRPC) patients. However, Enz-resistance inevitably emerges despite success of Enz in prolonging CRPC patients’ survival. Here we found that Enz-resistant prostate cancer (PCa) cells had higher BCL2 expression. We aimed to test whether targeting BCL2 would influence Enz sensitivity of prostate cancer (PCa) and identify the potential mechanism. Methods: The study was designed to target Enz-induced BCL2 with inhibitor ABT263 and test Enz sensitivity in Enz-resistant PCa cells by MTT assay. Cellular reactive oxygen species (ROS) levels were detected with dihydroethidium staining, and in vitro deubiquitinating enzyme activity assay was used to evaluate ubiquitin specific protease 26 (USP26) activity. Results: ABT263 could increase Enz sensitivity in both Enz-sensitive and Enz-resistant PCa cells via inducing ROS generation. Elevated cellular ROS levels might then inhibit USP26 activity to increase the ubiquitination of androgen receptor (AR) and AR splice variant 7 (ARv7) and their ubiquitin/proteasome-dependent degradation, which contributed to the increase of Enz sensitivity. In vivo mouse model also demonstrates that ABT263 will suppress the PCa progression. Conclusion: This study demonstrated that targeting Enz-induced BCL2 with inhibitor ABT263 could increase Enz sensitivity in both Enz-sensitive and Enz-resistant PCa cells through induction of cellular ROS levels and suppression of USP26 activity with a consequent increase of ubiquitin/proteasome-dependent degradation of AR and ARv7 protein expression.

## 1. Introduction

The incidence of prostate cancer (PCa) is the highest among common malignant tumors of men in the United States, and 164,690 new cases and 29,430 deaths were estimated in 2018 [1]. Elsewhere in the world, such as in China, although both incidence and mortality rates of PCa are lower, there has been a tremendous increase in the last few decades [2]. Besides surgical treatment, androgen deprivation therapy (ADT) with antiandrogens targeting androgens/androgen receptor (AR) signaling is the standard used to treat PCa in different clinical settings. ADT is commonly prescribed for men with high-risk localized PCa, recurrent, progressive, or metastatic PCa, all of which are still androgen sensitive [3]. However, most PCa patients will eventually develop cancer progression despite castrate levels of testosterone, into castration resistant prostate cancer (CRPC) [4]. Various second-generation antiandrogens, including Enzalutamide (Enz), have emerged and increased the numbers of options for treating CRPC patients [5,6]. However, even though Enz can extend patients survival for an extra 4.8 months, most of them will eventually develop Enz-resistance after 1–2 years [7].

Great effort has been made to identify potential mechanisms of Enz-resistance and many dysregulated pathways have been implicated. Androgen receptor (AR) splice variants, such as ARv7, are reported to be crucial in development of Enz-resistance [8] as ARv7 lacks the ligand binding domain that is targeted by Enz while still possessing transcriptional activation in the absence of androgen. Recent in-depth large scale genomic analysis of PCa indicated that AR is critical for PCa progression even at the stage of CRPC [9]. In addition, Enz-resistant (EnzR) PCa also had higher AR expression [10]. Thus, targeting AR/ARv7 likely will provide therapeutic efficacy including increasing Enz sensitivity. 

The ubiquitin-proteasome system controls a variety of cellular biological functions by mediating protein degradation, including AR [11]. Deubiquitinating enzymes (DUBs) are proteases that can cleave ubiquitin or ubiquitin-like proteins from target proteins to maintain the balance of ubiquitination dynamics [12]. Ubiquitin specific protease (USP) is one of the major members of the DUB family, and several USPs were reported to be associated with AR ubiquitination and degradation [13,14,15,16,17].

Here we found BCL2 expression increased after Enz treatment and in Enz resistant (EnzR) PCa cells. ABT263, a potent BCL2 family protein inhibitor [18], increased Enz sensitivity in both Enz-sensitive (EnzS) and EnzR PCa cells. It did so likely through induction of cellular reactive oxygen species (ROS) levels to suppress ubiquitin-protease with a consequent increase of ubiquitin/proteasome-dependent degradation of AR and ARv7 protein expression (See diagram in Appendix A). This preclinical study established the foundation to use ABT263 in combination with Enz to treat PCa patients, especially those resistant to Enz.

## 2. Materials and Methods

### 2.1. Cell Culture and Reagents

C4-2, CWR22Rv1, and HEK293T cell lines were purchased from the American Type Culture Collection (ATCC, Manassas, VA, USA) and tested for mycoplasma contamination. C4-2 and CWR22Rv1 cells were cultured in RPMI 1640 media, and HEK293T cells were cultured in DMEM media. Both media contained 1% penicillin and streptomycin, as well as 10% fetal bovine serum (FBS). C4-2 EnzR cell lines (EnzR1-C4-2) were generated via chronic culture of CRPC C4-2 cells (EnzS1-C4-2) in media containing increasing Enz concentrations (from 10 µM to 40 µM) for 3 months at each concentration. CWR22Rv1 cells are naturally resistant to Enz and were named as Enz-R3-CWR22Rv1 for these studies. All cells were maintained in a humidified 5% CO_2_ environment at 37 °C. All cell lines were authenticated and detected to be mycoplasma and bacteria free following ATCC’s instructions. Enz was used at 10 μM for EnzR1-C4-2 cell culture. ABT263 (Adooq, # A10022, Irvine, CA, USA) was used at 5 μM for EnzS1-C4-2 and EnzR1-C4-2 cells, and at 10 μM for EnzR3-CWR22Rv1 cells. Cycloheximide (CHX) (Adooq, # A10036) was used at 1 μg/mL, MG132 at 10 μM (Adooq, # A11043), Bortezomib at 0.5 μM (Selleckchem, #S1013, Houston, TX, USA), N-acetyl-cysteine (NAC) (Adooq, # A10032) was used at 3 mM, and Z-VAD-FMK at 10 μM (Adooq, # A12373).

### 2.2. Cell Proliferation Assays

We plated 5000 EnzS1-C4-2, EnzR1-C4-2 or EnzR3-CWR22Rv1 cells into each well of 48-well plates. We harvested cells on Day 0 and Day 4, followed with MTT assays. Briefly, 50 μL of 5 mg/mL MTT was added to each well and blank controls of media without cells. After incubating the plates for 2 h at 37 °C, media was removed and 150 μL DMSO added per well to dissolve the precipitate. The plates were then placed on an orbital shaker for 15 min covered with foil. The absorbance was measured at 570 nm. As for colony formation assay, we plated 500 EnzS1-C4-2, EnzR1-C4-2, or EnzR3-CWR22Rv1 cells into each well of 6-well plates. After two-week drug treatment, colonies were fixed with methanol, treated with Giemsa stain, and counted by using ImageJ software [19]. A colony is defined as a nonoverlapping group of at least 50 cells, as described previously [20].

### 2.3. Drug Synergy Assay

We performed drug synergy assay using Chou-Talalay method to calculate CI (combination index) value for determining the synergistic effect of ABT263 and Enz [21]. EnzS1-C4-2 and EnzR1-C4-2 cells were treated with serial dilutions of ABT263 and Enz alone or with combination in the same concentration ratio of 2:1 for 4 days. For EnzR3-CWR22Rv1 cells, the concentration ratio was 1:1. The MTT assay was performed to determine cell viability. Data was analyzed using CompuSyn software based on median-effect principle and combination index (CI) theorem (CI < 1, CI = 1, and CI > 1 indicates synergism, additive effect, and antagonism, respectively).

### 2.4. Lentivirus Packaging and Cell Transfection

The plasmids used in the study, including sequences, were described in Appendix A. The sh-USPs were constructed into the pLKO.1 lentiviral vector. The pLKO.1 sh-USPs (USP7, USP12, USP14, USP22, and USP26) together with package and envelope plasmids, psPAX2, and pMD2G, were co-transfected into HEK293T cells for 48 h following the standard calcium phosphate transfection method to produce the shRNA lentivirus particle soup, which was then collected, concentrated, and stored at −80 °C for infecting PCa cells later.

### 2.5. RNA Extraction and Quantitative Real-Time PCR (PCR) Analysis

Total RNAs were isolated using Trizol reagent (Invitrogen, Grand Island, NY, USA) and 2 μg of each total RNA was subjected to reverse transcription using Superscript III transcriptase (Invitrogen). Real-time PCR was conducted using a Bio-Rad CFX96 system with SYBR green to determine the mRNA expression level of a gene of interest. Expression levels were normalized to the control GAPDH (Glyceraldehyde 3-phosphate dehydrogenase) level using the 2^−ΔΔ^Ct method.

### 2.6. Western Blot and Protein Stability Analysis

Cells were washed twice with cold PBS and lysed in RIPA buffer and proteins (40–50 μg) were separated on 6–10% SDS/PAGE gel and then transferred onto PVDF (Polyvinylidene fluoride) membranes (Millipore, Billerica, MA, USA). After PVDF membranes were blocked with non-fat milk, they were sequentially incubated with primary antibodies, horseradish peroxidase (HRP)-conjugated secondary antibodies, and visualized using the enhanced chemiluminescence (ECL) system (Thermo Fisher Scientific, Rochester, NY, USA). The primary antibodies used in the study included the following, AR (Santa Cruz, #sc-816, Paso Robles, CA, USA), GAPDH (Santa Cruz, #sc-166574), α Tubulin (Santa Cruz, #sc-8035), BCL2 (Santa Cruz, #sc-783), ubiquitin (Santa Cruz, #sc-8017), PSA (ABclonal, # A2052, Woburn, MA), USP7 (ABclonal, # A2345), USP12 (ABclonal, # A13247), USP14 (Santa Cruz, #sc-398009), USP22 (Santa Cruz, #sc-390585), and USP26 (ABclonal, #A7999, Woburn, MA, USA). To analyze protein stability of AR and ARv7, equivalent numbers of cells were seeded and treated with DMSO or ABT263 for 48 h. Cycloheximide (CHX) at 1 μg/mL was added, and cells were harvested at 0 h, 3 h, 6 h, 9 h, and 12 h. For the interruption assays, 10 μM MG132 and 0.5 μM Bortezomib were added after 48 h of DMSO or ABT263 treatment. Cells were harvested after 4 h incubation for western blot. Intensity of western blot bands were quantified using Image Lab Software (Bio-rad, Hercules, CA, USA) and normalized to the control.

### 2.7. Immunoprecipitation (IP)

Cells were washed twice with cold PBS and lysed in lysis buffer (50 mM HEPES/pH 7.6, 300 mM NaCl, 1.5 mM MgCl_2_, 10% Glycerol, 1% Triton X-100, 1 mM EGTA, 0.1 mM EDTA, 20 mM NaF, 10 mM Na_4_P_2_O_7_, 1 mM Na_3_VO_4_, and 1 mM DTT protease inhibitor cocktail (Roche Applied Science, Penzberg, Germany) on ice for 30 min. Supernatants from cell lysis were collected after spinning at 12,000 RPM at 4 °C for 5 min. After detection of protein concentration, the lysates were incubated with 2 μg AR primary antibody and rotated at 4 °C overnight. Then 15 μL of Protein A/G–agarose beads (Thermo Fisher Scientific, Inc., Fremont, CA, USA) were added the next day for incubation by rotation at 4 °C for 1 h. The beads were washed with lysis buffer three times and boiled for 10 min with SDS loading buffer. Western blot was performed as described above for detection of ubiquitination.

### 2.8. Reactive oxygen species (ROS) Measurement

Cellular ROS levels were detected with dihydroethidium (DHE). Cells were washed with PBS and subsequently incubated with 10 μM DHE at 37 °C in the dark for 30 min. After that, cells were rinsed twice with PBS and were immediately observed under fluorescence microscope (Leica DMI6000B, Leica Microsystems GmbH, Mannheim, Germany). Photos were captured and representative images presented. We also used a BioTek fluorescent plate reader (Synergy™ Mx, BioTek Instruments, Inc., Winooski, VT, USA) and measured the fluorescence. Cells were seeded into 96-well black plates and incubated with 10 μM DHE at 37 °C in the dark for 30 min. Culture media with DHE were carefully aspirated off, and cells were then rinsed with PBS once. The 96-well black plates were placed on the fluorescent plate reader and we measured fluorescence using an excitation wavelength of 485 nm and an emission wavelength of 580 nm.

### 2.9. In Vitro Deubiquitinating Enzymes (DUB) Activity Assay

For measuring USPs activity, we used the chemically modified ubiquitin activity probe, HA-tag ubiquitin vinyl methyl ester (HA-UbVME) (Enzo, #BML-UW0880-0025, Farmingdale, NY, USA). DUB activity assay was performed according to the protocol described by Borodovsky et al. [22] with several modifications. Cells were lysed on ice for 1 h with buffer (250 mM Tris, 3 mM EDTA, 150 mM NaCl, and 0.5% NP40). Supernatants from cell lysates were collected after spinning at 14,000 RPM at 4 °C for 10 min. Cell extracts and 1 μg of HA-UbVME probe were incubated at 25 °C for 60 min in DUB reaction buffer (50 mM Tris, 1 mM EDTA, 50 mM NaCl, 10% glycerol). Reactions were terminated with SDS loading buffer, boiled for 10 min, and processed for Western blot analysis.

### 2.10. In Vivo Studies

For in vivo tumor growth studies, 16 male BALB/c nude mice were randomly divided into four groups (n = 4 per group). Based on the in vitro study, four mice each group was estimated to be enough to observe the potential synergistic effect. EnzR3-CWR22Rv1 cells were harvested, washed, re-suspended in serum-free media, and mixed 1:1 with Matrigel, then injected subcutaneously into the flank of nude mice (1 × 10^7^ cells/mouse). After 4–6 weeks of tumor growth, the mice were intraperitoneally treated with vehicle control, 30 mg/kg Enz, 50 mg/kg ABT263, or the combination of Enz plus ABT263 every 2 days for 2 weeks without blinding. Tumors were measured weekly by caliper. After another 2–3 weeks, mice were sacrificed and tumors were subjected to immunohistochemistry (IHC) studies. If any mouse died before tumor harvest, it would be excluded from analysis. Protocols for animal care and experimentation were approved by the University Committee on Animal Resources (UCAR) of the University of Rochester Medical Center (UCAR 2002-296E).

### 2.11. Immunohistochemistry (IHC) Staining

IHC was performed on the samples from mouse xenografted tumors. Briefly, the samples were fixed in 4% neutral buffered paraformaldehyde for 18 h, then embedded in paraffin and cut into 4 μm slices. After deparaffinization, hydration, antigen retrieval, and blocking, these sections were incubated with corresponding primary antibodies, incubated with biotinylated secondary antibodies (Vector Laboratories, Burlingame, CA, USA), and then visualized by VECTASTAIN ABC peroxidase system and 3, 3′-diaminobenzidine (DAB) kit (Vector Laboratories, Burlingame, CA, USA). The slides were reviewed separately by two experienced pathologists. Average optical density of staining was calculated using ImageJ software [19].

### 2.12. Statistics

Experiments were repeated independently at least 3 times with triplicate data points. Results are expressed as mean ± SD. The Student’s *t*-test and two-way ANOVA test were applied to determine statistical significance with SPSS 22 (IBM Corp., Armonk, NY, USA) or GraphPad Prism 6 (GraphPad Software, Inc., La Jolla, CA, USA). A *P*-value of less than 0.05 was considered statistically significant.

## 3. Results

### 3.1. BCL2 Expression Increased in Enz-Resistant Prostate Cancer (PCa) Cells

As mis-regulation of cellular apoptosis is critical for the tumor development, particularly for hematopoietic cancers [23], we first examined if altered cell apoptosis might also play a role in PCa development. The results from the GEO dataset [24,25] revealed that the mRNA expression of BCL2, a key player in the PCa cell apoptosis increased dramatically in both CRPC cell-derived tumors (LAPC9 androgen independent cells and LNCaP secondary CRPC cells) (Figure 1A) [24] and EnzR cell-derived tumors (CWR-R1, LAPC-4, LNCaP Enz-resistant cell) (Figure 1B) [25]. Treating with 10 μM Enz also increases both BCL2 mRNA (Appendix A) and protein (Figure 1C) expression in the PCa EnzS1-C4-2 cells. We also found higher BCL2 expression in the PCa EnzR1-C4-2 cells as compared to their parental EnzS1-C4-2 cells (Figure 1D). 

Together, results from Figure 1A–D and Appendix A suggest that Enz treatment may increase BCL2 expression, and higher BCL2 expression is found in the EnzR PCa cells. 

### 3.2. BCL2 Targeting by ABT263 Impedes Cell Growth by Enhancing Enz Potency in Both Enz-Sensitive and Enz-Resistant PCa Cells

To test the potential linkage of Enz-increased BCL2 to the development of Enz-resistance, we used ABT263 (Navitoclax), a BCL2 inhibitor, to determine if it would influence Enz sensitivity in the PCa cells. Results from MTT proliferation assay revealed that treating with ABT263 could increase Enz sensitivity (Figure 1E–G) in EnzS1-C4-2 cells (Figure 1E, 54.9% vs. 37.4% decrease of cell viability, *p* = 0.003), EnzR1-C4-2 (Figure 1F, 47.9% vs.24.0% decrease of cell viability, *p* = 0.018) and EnzR3-CWR22Rv1 cells (Figure 1G, 11.9% vs. 5.3% decrease of cell viability, *p* = 0.046). We also performed colony formation assay to confirm this finding (representative data was shown in Appendix A). 

To confirm the synergistic effect of ABT263 and Enz, we performed drug synergy assay and calculated CI value, and found that ABT263 and Enz had synergistic effects to suppress cell growth of EnzS1-C4-2, EnzR1-C4-2, and EnzR3-CWR22Rv1 cells (Figure 1H). Besides, ABT263 would not decrease both mRNA (Appendix A) and protein (Appendix A) expression of BCL2.

Together, results from Figure 1A–H and Appendix A suggest that targeting BCL2 with ABT263 can increase Enz sensitivity to further suppress both EnzR and EnzS PCa cell growth.

### 3.3. ABT263 Mechanistically Increases Enz Sensitivity by Enhancing Proteasome-Dependent Degradation of AR and ARv7

To dissect the mechanism underlying ABT263-increased Enz sensitivity, we focused on the ARv7, as a recent clinical study clearly indicated that EnzR PCa patients have higher ARv7 expression and Enz treatment could increase the ARv7 expression in their PCa tumors [26]. Results from western blot assays revealed that treating with ABT263 led to decrease AR protein expression in EnzS1-C4-2 cells, as well as AR and ARv7 in EnzR1-C4-2 and EnzR3-CWR22Rv1 cells (Figure 2A). 

Interestingly, treating with ABT263 failed to significantly decrease the mRNA expression of AR in the EnzS1-C4-2 cells as well as the mRNA expressions of AR and ARv7 in EnzR1-C4-2 and EnzR3-CWR22Rv1 cells (Figure 2B), suggesting that ABT263 may suppress the AR and ARv7 expression mainly at the protein level. Expression of AR target genes after ABT263 with or without Enz treatment in EnzS1-C4-2 cells was also tested. As shown in the Appendix A, greater suppression of PSA and FKBP5 was observed with combination treatment of ABT263 and Enz than ABT263 or Enz treatment alone. 

We then assayed ABT263 impact on the protein stability via treating with cycloheximide (CHX), and results revealed that ABT263 decreased the stability of AR in the EnzS1-C4-2 cells (Figure 2C) as well as the stability of AR and ARv7 in the EnzR1-C4-2 (Figure 2D) and EnzR3-CWR22Rv1 (Figure 2E) cells. Line graphs of AR/ARv7 protein stability were drawn after repeating the western blot data of Figure 2C–E another two times with different samples, showing similar decreasing trends of AR/ARv7 stability in EnzS1-C4-2 (Appendix A), EnzR1-C4-2 (Appendix A), and EnzR3-CWR22Rv1 (Appendix A) cells after ABT263 treatment. We also over-expressed AR or ARv7 in EnzS1-C4-2 cells to test its significance as a key target for the ABT263 effect. As shown in Figure 2F,G, over-expression of AR or ARv7 decreased Enz sensitivity (8.4% and 8.7% respectively when treated with 15 μM Enz) as well as partly reversed ABT263 effect in sensitizing PCa to Enz. In order to further confirm that ABT263 is mainly through mediating the degradation of AR and ARv7 in both Enz-sensitive and Enz-resistant PCa cell lines, we tested ABT263 effect in PC3 cells (AR negative). As shown in Appendix A, ABT263 slightly decreases PC3 cell viability (left figure). However, after being normalized with the DMSO group, little increase of Enzalutamide sensitivity was observed, with no statistical significance (right figure). These data suggest that AR and ARv7 are the critical molecules targeted by ABT263.

As both AR and ARv7 could be degraded through the ubiquitin-proteasome pathway [27], we then applied the interruption assay to test whether blocking proteasome activity could reverse ABT263 effect. We treated with two proteasome inhibitors, MG132 and Bortezomib, and found that both could partly block the decrease of AR and ARv7 expression in EnzS1-C4-2, EnzR1-C4-2, and EnzR3-CWR22Rv1 cells (Figure 2H–J). Consistent with ubiquitin involvement, we also observed a dramatic increase in ubiquitination of AR after ABT263 treatment in both EnzS1-C4-2 (Figure 2K,L) and EnzR1-C4-2 cells (Figure 2M,N).

Together, results from Figure 2A–N and Appendix A suggest that targeting the BCL2 with ABT263 could decrease AR and ARv7 protein expression through increasing degradation via ubiquitin proteasome pathway.

### 3.4. ABT263 Mechanistically Increases AR and ARv7 Degradation by Inducing Cellular ROS Level

As BCL2 is a classical anti-apoptosis protein, we then tested the potential linkage of its anti-apoptotic activity to its ability to alter AR and ARv7 expression. The results revealed that inhibition of BCL2 activity by ABT263 (with and without Enz treatment) resulted in cleaved-PARP expression in EnzS1-C4-2, EnzR1-C4-2, and EnzR3-CWR22Rv1 PCa cells, suggesting that ABT263 could promote PCa apoptosis (Appendix A). However, blocking apoptosis through inhibition of caspase activity with Z-VAD-FMK (pan-caspase inhibitor) failed to reverse the decrease of AR and ARv7 protein expression (Appendix A), suggesting that the apoptotic pathway was unlikely responsible for the decrease of AR and ARv7 expression. 

Consistent with this, we found that the increased Enz sensitivity could not be blocked by Z-VAD-FMK in EnzS1-C4-2, EnzR1-C4-2, or Enz3-CWR22Rv1 PCa cells (Appendix A). As BCL2 is a mitochondria protein that may alter the mitochondria function via modulating the ROS [28], we then tested whether ABT263 could function via altering cellular ROS to decrease AR and ARv7 expression. We applied DHE staining to determine the cellular ROS level [29] and found that ABT263 treatment could significantly induce cellular ROS levels in EnzS1-C4-2, EnzR1-C4-2, and EnzR3-CWR22Rv1 PCa cells (Figure 3A–D). Importantly, adding N-acetyl-cysteine (NAC), a substrate of cellular glutathione system [30], could reverse the ABT263-increased ROS level in these three PCa cell lines (Appendix A) and reverse the Enz-sensitivity increase in EnzS1-C4-2, EnzR1-C4-2, and EnzR3-CWR22Rv1 PCa cells using MTT proliferation assays (Figure 3E–G).

Consistent with ROS’s role in altering the Enz-sensitivity, western blot assays showed that NAC could reverse the ABT263 decreased AR expression in EnzS1-C4-2 cells by (Figure 4A). NAC could also reverse the decrease of AR and ARv7 expression in EnzR1-C4-2 and EnzR3-CWR22RV1 cells by ABT263 (Figure 4B,C). In addition, the increase in ubiquitination of AR after ABT263 treatment in both EnzS1-C4-2 (Figure 4D,E and Appendix A) and EnzR1-C4-2 cells (Figure 4F,G) could be partly reversed by inhibition of ROS.

Together, results from Figure 3A–F, Figure 4A–G, Appendix A suggested that ABT263 could increase AR and ARv7 ubiquitination and degradation via inducing cellular ROS level in PCa cells.

### 3.5. ABT263 Increases Enz Sensitivity and AR/ARv7 Degradation via Inhibiting USP26 Activity

Protein ubiquitination is a balance between enzyme-mediated ubiquitination and de-ubiquitination. As deubiquitin enzymes (DUB) contain an active site of cysteine that is sensitive to cellular redox state, it is likely that ROS may impact the DUB enzymes to control AR/ARv7 protein levels. We applied in vitro DUB activity assay in both EnzS1-C4-2 cells and EnzR1-C4-2 and EnzR3-CWR22Rv1 cells and western blot assays, which showed that ABT263 could inhibit DUBs activity extensively (Figure 5A). 

To further dissect which DUB may play the main role in regulating AR and ARv7 degradation after ABT263 treatment, we screened five potential candidates, including USP7, USP12, USP14, USP22, and USP26, which had been reported to be involved in deubiquitinating AR [13,14,15,16,17]. We first knocked down these five USPs separately in the three PCa cell lines, and found that several candidates (USP14, USP22, and USP26) were shown to be able to regulate AR/ARv7 expression and reverse ABT263 effects (Figure 5B–D). However, USP26 is the only consistent one among all the three cell lines that could not only decrease AR and/or ARv7 expression, but also partly reverse ABT263 effect in all three PCa cell lines. We also confirmed the knock down efficiency by using western blot assays (Appendix A, Figure 5E–G, left panels).

To determine the critical role of USP26 in Enz sensitivity in PCa cells, we applied MTT proliferation assay in response to Enz treatment and found that knocking down USP26 itself could not only increase Enz sensitivity (9.8% in EnzS1-C4-2, 10.6% in EnzR1-C4-2, and 13.2% in EnzR3-CWR22Rv1 cells), but also partly reverse the sensitivity increase by ABT263 (Figure 5E–G, right panels) in EnzS1-C4-2 (E), EnzR1-C4-2 (F), and EnzR3-CWR22Rv1 (G) cells.

Consistent with this, knocking down USP26 could decrease AR and/or ARv7 protein stability in EnzS1-C4-2, EnzR1-C4-2, and EnzR3-CWR22Rv1 PCa cells (Figure 6A–C), likely due to increasing ubiquitination of AR in EnzS1-C4-2 (Figure 6D,E) and EnzR1-C4-2 cells (Figure 6F,G). Line graphs of AR/ARv7 protein stability were drawn after repeating the western blot data in Figure 6A–C another two times with different samples, showing similar decreasing trend of AR/ARv7 stability in EnzS1-C4-2 (Appendix A), EnzR1-C4-2 (Appendix A), and EnzR3-CWR22Rv1 (Appendix A) cells after knocking down USP26.

An in vitro DUB assay was applied to further detect USP26 activity, and results revealed that ABT263 could significantly decrease USP26 activity, which could be reversed by inhibition of cellular ROS (Figure 6H–J).

Together, results from Figure 5A–G, Figure 6A–J, Appendix A suggest that ABT263 may function via increasing the cellular ROS level to decrease USP26 activity to increase AR and ARv7 protein degradation, as well as Enz sensitivity in PCa cells.

### 3.6. Preclinical Study Using in Vivo Mouse Model to Demonstrate ABT263 Can Increase Enz Sensitivity in Enz-Resistant PCa Cells and Better Suppress of PCa Progression

Finally, we used the in vivo mouse model to study the synergistic effect of Enz and ABT263 in treating EnzR PCa cells. We first established subcutaneous transplanted EnzR3-CWR22Rv1 cell line model in nude mice. After 4–6 weeks, mice with palpable tumors (5 × 5 mm^2^) were divided into four groups and treated with DMSO, Enz, ABT263, or Enz plus ABT263. The subcutaneous tumor growth was monitored/measured weekly by caliper. We analyzed the data and found that the combination treatment of Enz plus ABT263 could significantly decrease tumor growth and achieve greater tumor suppression than treatment with either Enz or ABT263 alone (Figure 7A–B). Mice weights were comparable between groups and 50 mg/kg ABT263 treatment would not lead to dramatic weight loss (Figure 7C).

Two to three weeks after 10 injection treatments, we collected the subcutaneous tumor samples to detect AR and ARv7 in each group using immunostaining. AR and ARv7 staining were reduced in ABT263 alone group and Enz plus ABT263 group compared with vehicle group (Figure 7D,E), which is consistent with our in vitro results. Staining of AR target gene, PSA, was also reduced after ABT263 treatment (Figure 7F).

Together, results from our in vivo animal model in Figure 7A–F demonstrated that Enz with ABT263 could lead to better suppression of EnzR PCa cell growth.

## 4. Discussion

Early studies indicated that the expression of the BCL2 family is associated with development of androgen independent PCa [24,31,32,33]. BCL2 expression can also be induced by Enz treatment [24,25] and higher BCL2 expression was found in the EnzR patient derived xenografts [34]. This indicates that BCL2 is one of the potential therapeutic targets of EnzR PCa. ABT263, also known as Navitoclax, is an orally active anti-cancer drug, that disrupts BCL2/BCL-XL interactions [18] and has been evaluated in several clinical trials of chronic lymphocytic leukemia, non-small cell lung cancer, ovarian cancer, and some other solid tumors [35,36]. In conjunction with a recent study that discovered that another BCL2 inhibitor, ABT199, can be used in combination with Enz for the treatment of EnzR CRPC [24], we further dissected the ABT263 molecular mechanisms and showed here that ABT263 can decrease AR/ARv7 protein expression to increase Enz sensitivity of both EnzS and EnzR PCa cells.

BCL2 is a classic anti-apoptotic regulator [37] that is reported to be closely associated with resistance to chemotherapy [38,39]. BCL2 inhibitors, including ABT263, can also increase PCa chemo-sensitivity to docetaxel [40] and paclitaxel [41] through enhancing chemotherapy induced apoptosis. In addition, another BCL2 inhibitor, HA14-1, can also overcome the radiation therapy resistance of PCa cells through increasing ROS generation and c-Jun N-terminal kinase (JNK) activation [42]. 

We showed that ABT263 can increase cellular ROS levels, and inhibition of ROS can reverse the decrease of AR/ARv7 protein expression as well as the increase of Enz sensitivity. These findings are consistent with several other studies showing that ABT263 alone [43] or in combination with other drugs [44] can induce the generation of ROS. Although increased ROS generation is often observed in treatment with many anticancer drugs and natural compounds [45,46], it can’t be taken for granted that all ROS generation is identical since ROS consists of several different chemically reactive species that contain various compounds of oxygen, such as peroxides, superoxides, hydroxyl radicals, singlet oxygen, alpha-oxygen, etc. [47]. The induced ROS generation by ABT263 in this study was found to be responsible for the increase of both AR/ARv7 protein degradation and Enz sensitivity. However, it is not clear, or most likely not the case, that any ROS-inducing drug or compound would affect AR/ARv7 protein degradation as well as Enz sensitivity of PCa cells/tissues as the magnitude and specificity of a particular ROS species might play a role in a particular regulation of cellular processes. 

The DUBs are a large group of proteases that cleave ubiquitin from its substrate proteins and other molecules [12] and maintain the balance of ubiquitination dynamics and regulate protein degradation [48]. The largest family of DUBs, the ubiquitin specific proteases (USP/UBP), are cysteine-dependent and consist of 56 members. The activity of USPs are relatively sensitive to ROS and oxidative stress that can inactivate USPs catalytic function [49]. In our current study, we screened several AR related USPs, including USP7, USP12, USP14, USP22, and USP26 and found that the USP26 activity was repressed by the induced cellular ROS, which is responsible for AR/ARv7 protein degradation and Enz sensitivity increase by ABT263. 

Besides USP26, USP22 was also found to regulate AR splice variants (AR-Vs). Knocking down USP22 expression or inhibiting its activity can dramatically reduce AR-Vs expression [15]. In our study, we also found some other USPs candidates (USP14, USP22) could decrease AR/ARv7 expression, besides USP26, although they can’t consistently reverse the ABT263 effect in all the three cell lines. USPs may be potential therapeutic targets to control CRPC transition and to treat CRPC by reducing the heretofore non-targetable AR-Vs. However, no compounds targeting DUBs, including USPs, have been approved for clinical use so far. USPs are likely to be druggable due to the defined catalytic residues. Several potential USPs inhibitors have shown some promising and exciting pre-clinical effect [50,51,52,53]. 

Based on the promising therapeutic effect of targeting BCL2 in treating PCa, especially CRPC, two registered clinical trials have been conducted so far. One study (NCT01828476) focused on ABT263 combined with Abiraterone acetate in treating patients with progressive metastatic castrate refractory prostate cancer (mCRPC). However, this study was terminated in 2016 due to slow accrual. The other study was using another BCL2 inhibitor, ABT199, combined with Enz in treating patients with mCRPC (NCT03751436). This study started in June 2019, and is still recruiting patients now. Although no results have been reported yet, we are looking forward to seeing that targeting BCL2 would provide promising therapeutic effect in treating CRPC soon.

## 5. Conclusions

In summary, our study reveals a promising therapeutic option to increase Enz sensitivity in PCa by ABT263 treatment, which targets BCL2. The increased ROS levels generated by ABT263 can inhibit USP26 activity to ubiquitinate AR/ARv7, and further sensitize PCa cells to Enz treatment (Appendix A). These findings suggest BCL2 and USP26 might be potential therapeutic targets for PCa patients. Furthermore, this study lays the basic foundation evidence for future clinical use of ABT263 in increasing Enz sensitivity.

## Figures and Tables

**Figure 1 cancers-12-00831-f001:**
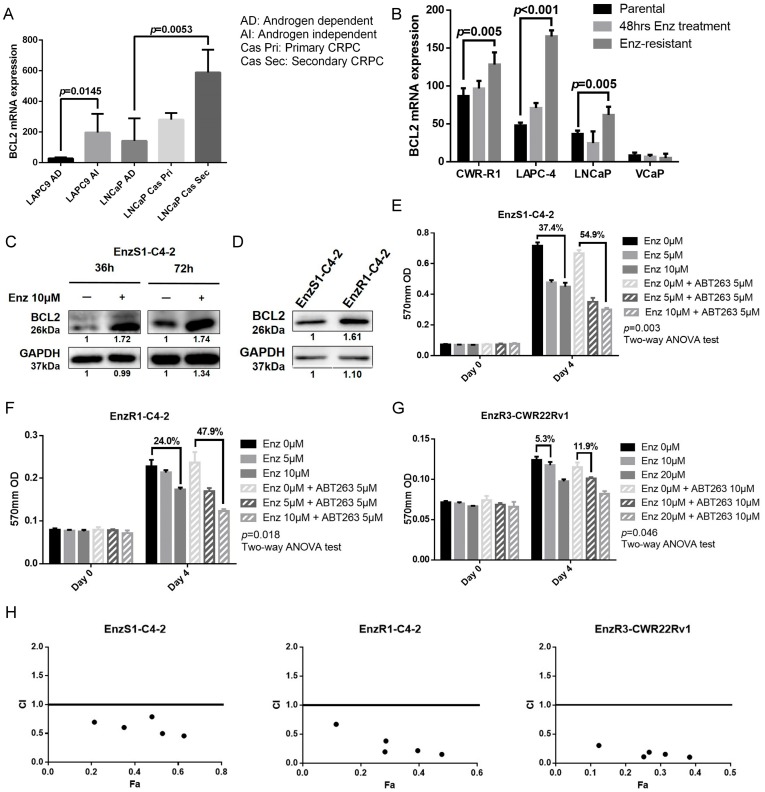
Targeting BCL2 increases Enz-sensitivity. (**A**) BCL2 mRNA is increased in castration-resistant PCa cell lines. BCL2 (ENSG00000171791.11) mRNA expression from GEO dataset (GSE88752) was analyzed for androgen-dependent (AD) and androgen-independent (AI) LAPC9 cell-derived tumors, as well as AD LNCaP, LNCaP Primary castration resistant prostate cancer (CRPC) (castration resistant, Cas Pri) and LNCaP Secondary CRPC (Enz resistant, Cas Sec) cell-derived tumors. (**B**–**D**) Enz treatment increases BCL2 expression. (**B**) BCL2 mRNA expression from GEO dataset (GSE78201) was analyzed for cells with Enz treatment for 48 h, and Enz-resistant cells. Two transcripts of BCL2 (NM_000633.2 and NM_000657.2) were measured, but NM_000657.2 was excluded for huge variation. Data of NM_000633.2 is shown in B. BCL2 protein expression in response to Enz treatment for 36 h and 72 h (**C**), and BCL2 protein expression in EnzS1-C4-2 and EnzR1-C4-2 cells (**D**). (**E**–**G**) Targeting BCL2 with ABT263 enhances sensitivity for Enzalutamide in EnzS1-C4-2 cells (**E**), in EnzR1-C4-2 cells (**F**) and EnzR3-CWR22Rv1 cells (**G**). Two-way ANOVA test was applied for determining the significant interaction between Enz and ABT263. (**H**) We used CompuSyn software based on median-effect principle and combination index (CI) theorem (CI < 1, CI = 1, and CI > 1 indicates synergism, additive effect and antagonism, respectively). This confirmed that ABT263 and Enz had synergistic effects to suppress cell growth of EnzS1-C4-2, EnzR1-C4-2, and EnzR3-CWR22Rv1 cells. Data are presented as Mean ± SD. #: P-value for interaction effect between Enz and ABT263 using two-way ANOVA.

**Figure 2 cancers-12-00831-f002:**
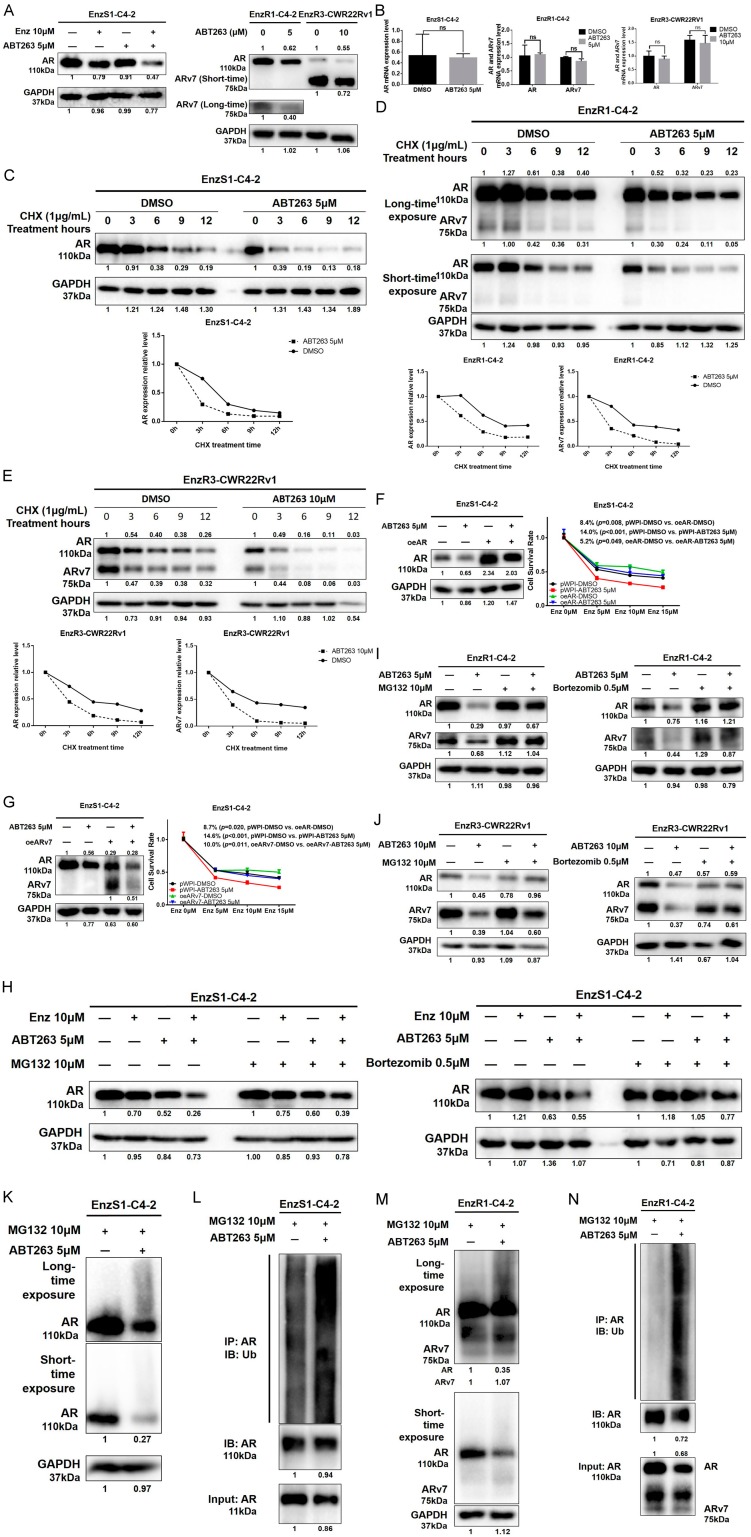
ABT263 increases ubiquitin-proteasome-dependent degradation of AR and ARv7. (**A**) ABT263 decreases AR and ARv7 protein expression. EnzS1-C4-2, EnzR1-C4-2, and EnzR3-CWR22Rv1 cells were treated with ABT263 or DMSO for 48 h. Chemiluminescence on the western blot was detected with short and long length of time to determine AR and ARv7 protein expression. (**B**) ABT263 does not alter AR and ARv7 mRNA expression. EnzS1-C4-2, EnzR1-C4-2, and EnzR3-CWR22Rv1 cells were treated with ABT263 or DMSO for 48 h. Q-PCR assay was applied to measure AR and ARv7 mRNA expression. (**C–E**) ABT263 decreases AR and ARv7 protein stability. Cycloheximide was used to measure the metabolic stability of AR and ARv7 in EnzS1-C4-2 cells (**C**), in EnzR1-C4-2 cells (**D**) and in EnzR3-CWR22Rv1 cells (**E**). Chemiluminescence on the western blot was detected with short and long length of time, are shown in (**D**). Note that ARv7 is more visible with longer exposure time. (**F**,**G**) Over-expressing AR or ARv7 partly reverses the increase of Enz sensitivity by ABT263. EnzS1-C4-2 cells were infected with pWPI, oeAR, or oeARv7 virus and treated with ABT263 5μM or DMSO. MTT proliferation assay was applied at Day 4 to measure cell proliferation (**F**, right panel; **G**, right panel). Western blot assay was used to confirm the efficiency of over-expressing AR (**F**, left panel) or ARv7 (**G**, left panel). (**H**–**J**) Proteasome inhibitors (MG132 and Bortezomib) partly block the decrease of AR and ARv7 caused by ABT263 in EnzS1-C4-2 cells (**H**), in EnzR1-C4-2 cells (**I**) and in EnzR3-CWR22Rv1 cells (**J**). (**K**–**N**) ABT263 increases ubiquitination of AR and ARv7. Increased slower mobility species of AR in EnzS1-C4-2 and EnzR1-C4-2 cells upon ABT263 treatment (**K**,**M**). Two exposures, short and long time, are shown. Note that the slower mobility species of AR is more visible with longer exposure time. Increased AR ubiquitination with ABT263 treatment (**L**,**N**). Data are presented as Mean ± SD. ns: not significant.

**Figure 3 cancers-12-00831-f003:**
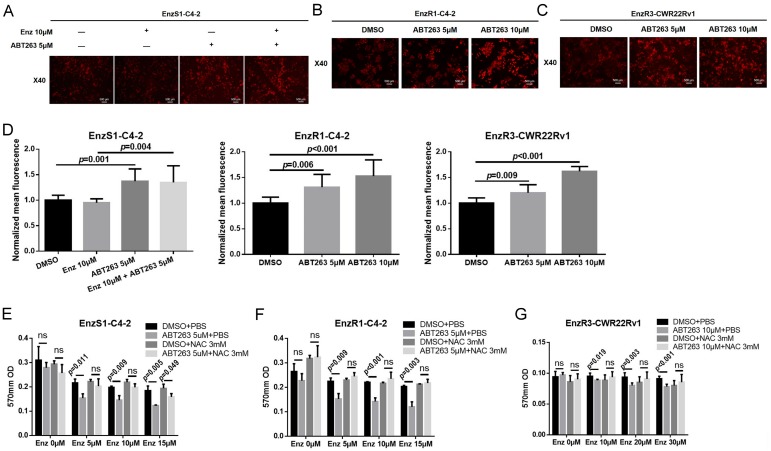
ABT263 increases Enz sensitivity through inducing cellular ROS. (**A**–**D**) ABT263 induces cellular ROS. Dihydroethidium (DHE) staining was applied to detect cellular ROS level in EnzS1-C4-2 cells (**A**), EnzR1-C4-2 cells (**B**) and EnzR3-CWR22Rv1 cells (**C**) after 48-hour ABT263 treatment. Fluorescence (excitation wavelength 485 nm and an emission wavelength 580 nm) was measured to quantify cellular ROS level and it also showed increased ROS level after ABT263 treatment (**D**). (**E**–**G**) ROS suppressor (N-acetyl-cysteine, NAC) could reverse the Enz-sensitivity increase by ABT263. MTT proliferation assay was used to detect cell proliferation in EnzS1-C4-2 cells (**E**), EnzR1-C4-2 cells (**F**), and EnzR3-CWR22Rv1 cells (**G**). Data are presented as Mean ± SD. ns: not significant.

**Figure 4 cancers-12-00831-f004:**
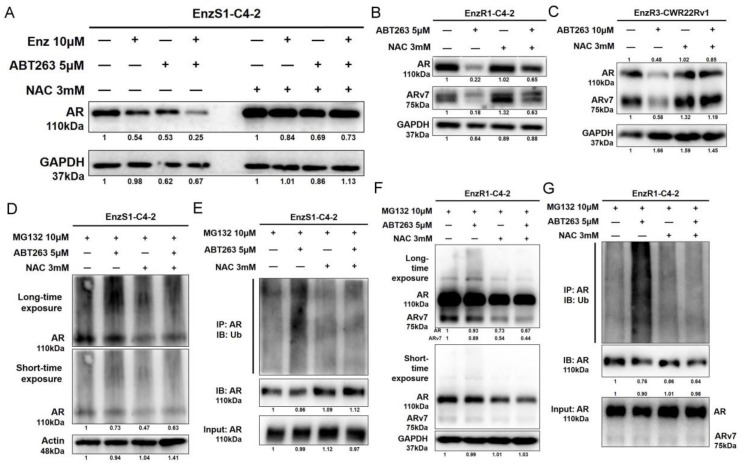
ABT263 increases AR/ARv7 degradation through inducing cellular ROS. (**A**–**C**) NAC reverses the ABT263 decrease of AR and ARv7 protein expression. Western blot assay was applied to measure AR and ARv7 expression in EnzS1-C4-2 cells (**A**), EnzR1-C4-2 cells (**B**), and EnzR3-CWR22Rv1 cells (**C**). (**D**–**G**) NAC reverses the ABT263 increased ubiquitination of AR and ARv7. Increased slower mobility species of AR in EnzS1-C4-2 and EnzR1-C4-2 cells upon ABT263 treatment could be reversed by NAC (**D**,**F**). Two exposures, short and long time, are shown. Note that the slower mobility species of AR is more visible with longer exposure time. Increased AR ubiquitination upon ABT263 treatment could also be reversed by NAC (**E**,**G**). Data are presented as Mean ± SD.

**Figure 5 cancers-12-00831-f005:**
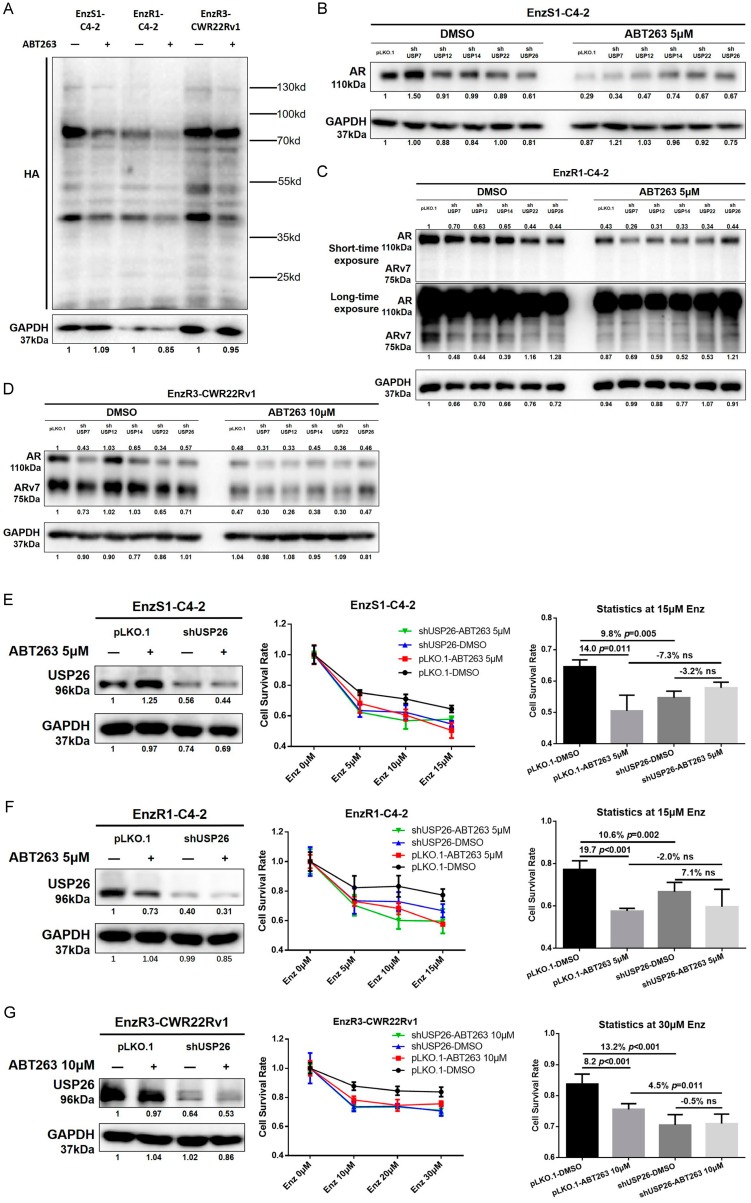
ABT263-induced cellular ROS inhibits USP26 activity leading to increase of Enz sensitivity in PCa cells. (**A**) ABT263 inhibits general deubiquitinating enzymes (DUB) activity. In vitro DUB activity assays and western blot assays were used to detect general DUB activity in EnzS1-C4-2, EnzR1-C4-2, and EnzR3-CWR22Rv1 cells. Active DUB protein could be distinguished from the inactive species by immunoblotting with anti-HA antibody. (**B–D**) ABT263 decreases AR and ARv7 protein expression mainly through USP26. We used lentivirus construction to knock down five ubiquitin specific protease (USP) candidates. Western blot assay was used to detect which USP, upon knock down, could not only decrease AR and ARv7 expression, but also block ABT263 effect in EnzS1-C4-2 cells (**B**), EnzR1-C4-2 cells (**C**), and EnzR3-CWR22Rv1 cells (**D**). Two exposures, short and long time, are shown in **C**. Note that ARv7 is more visible with the longer exposure time. (**E**–**G**) Knocking down USP26 reverses the increase of Enz-sensitivity by ABT263. MTT proliferation assay was used to detect cell proliferation in EnzS1-C4-2 cells (**E**), EnzR1-C4-2 cells (**F**), and EnzR3-CWR22Rv1 cells (**G**). Data are presented as Mean ± SD. ns: not significant.

**Figure 6 cancers-12-00831-f006:**
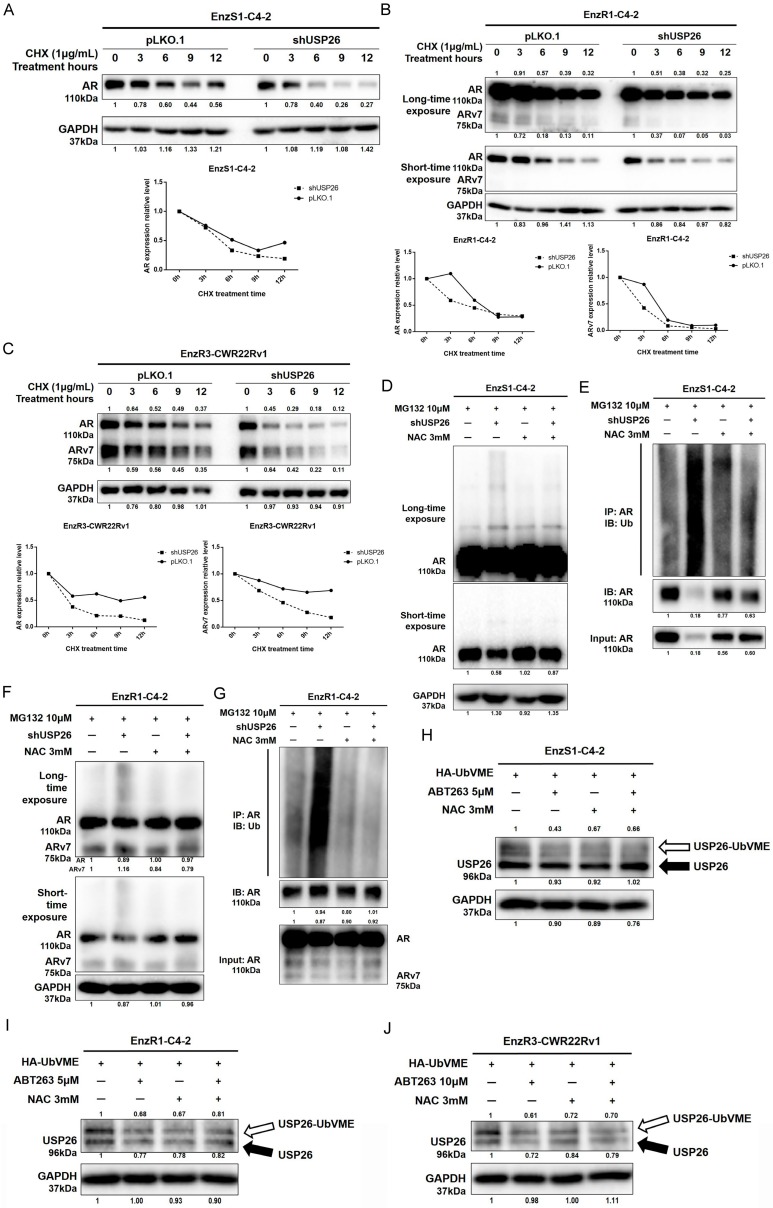
ABT263-induced cellular ROS inhibits USP26 activity leading to degradation of AR/ARv7. (**A–C**) USP26 contributes to protein stability of AR and ARv7. Cycloheximide (CHX) was used to measure the stability of AR and ARv7 in EnzS1-C4-2 cells (**A**), EnzR1-C4-2 cells (**B**) and EnzR3-CWR22Rv1 cells (**C**) after knocking down USP26. Two exposures, short and long time, are shown in (**F**). Note that ARv7 is more visible with longer exposure time. (**D–G**) NAC abolishes the increase of ubiquitination of AR and ARv7 by knocking down USP26. Increased slower mobility species of AR in EnzS1-C4-2 and EnzR1-C4-2 cells by knocking down USP26 could be inhibited by NAC (**D**,**E**). Two exposures, short and long time, are shown. Note slower mobility species of AR is more visible with longer exposure time. Increased AR ubiquitination upon ABT263 treatment could also be prevented by NAC (**F**,**G**). (**H**–**J**) NAC abolishes ABT263-inhibited USP26 activity. In vitro DUB assay and western blot assay were used to detect USP26 activity in EnzS1-C4-2 cells (**H**), EnzR1-C4-2 cells (**I**), and EnzR3-CWR22Rv1 cells (**J**). Active USP26 can bind to HA-UbVME probe thus being distinguished from the inactive probe through increased molecular weight. Data are presented as Mean ± SD.

**Figure 7 cancers-12-00831-f007:**
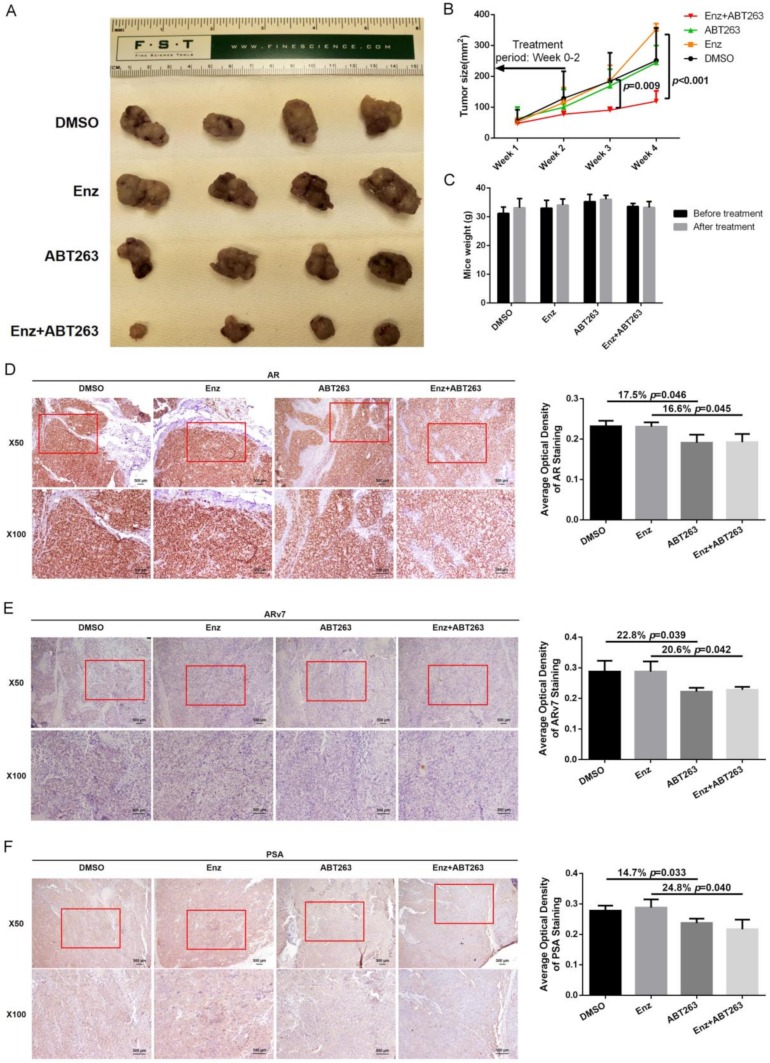
Preclinical study using in vivo mouse model to demonstrate ABT263 increases Enz sensitivity in Enz-resistant PCa cells (EnzR3-CWR22Rv1) and better suppress of PCa progression. (**A**,**B**) ABT263 increases Enz sensitivity and decreases xenograft tumor size. Xenograft tumors from mice after 4–5 weeks’ drug treatment were displayed (**A**). Tumor size was measured every week after drug delivery and reduction of the xenografts was observed in Enz with ABT263 treatment group (**B**). (**C**) Mice weight before and after drug treatment was comparable; (**D**–**F**) ABT263 decreases AR, ARv7, and PSA expression in xenograft tumors. Representative immunohistochemistry (IHC) images of AR in xenograft tumors (**D**), of ARv7 in xenograft tumors (**E**), and of PSA in xenograft tumors (**F**). Average optical density of staining was calculated using ImageJ software. Data are presented as Mean ± SD.

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
