# Peer review of "Preclinical Study Using ABT263 to Increase Enzalutamide Sensitivity to Suppress Prostate Cancer Progression Via Targeting BCL2/ROS/USP26 Axis Through Altering ARv7 Protein Degradation"

_cancers, 2020, doi:10.3390/cancers12040831_

Round 1

Reviewer 1 Report

In this study, Xu and colleagues investigated the effect of ABT263 on inhibiting BCL2 and enzalutamide resistant prostate cancer and found it was effective in both enz sensitive and resistant cells. They further observed increased reactive oxygen species when treated the cells with ABT263, which might inhibit USP26 and thus increase the ubiquitination and degradation of AR and ARv7. Overall, this study provided a novel angel which connects enz resistance, BCL2 and USP26 mediated AR degradation, and potentially a new therapeutic option for enz resistant prostate cancer patients.  However, I do have several concerns and suggestions about this manuscript.

  1. It is well documented that BCL2 and its mediated signaling pathways are increased in enzalutamide resistant prostate cancer, and idea of combined therapies of enz+BCL2 pathway inhibitors have been tested as well. for example, in Piling et al. 2019 (PMC6617752), Luk et al, 2017 (PMID: 27663589), Chen et al., 2019 (PMID: 31582422). Thus, this part of the study is not very novel.

  1. The authors have examined the tumor cell survival majorly through MTT assay, which has a lot of limitations when measuring cell viability (high toxicity, low sensitivity, reflect metabolism not real cell number et al), especially when enz treatment also influent PCa cell metabolic status. In contrast, the authors should use ATP-based luminescent assay, such as CellTiter-Glo or RealTime-Glo type of assay to measure PCa cell proliferation.

  1. Beside apoptosis, enz is known to significantly slow down the cell cycle and cause cell cycle arrest. The author should measure apoptotic molecules such as cleaved caspase-3, as well as cell cycle by measuring BrdU or Edu incorporation to show that the effect of ABT263 is really through the regulation of apoptosis.

  1. The authors proposed that the effect of ABT263 is mainly through the degradation of AR/AR-v7, in both enz sensitive and enz resistant PCa cell lines. If this is the case, then this reagent should have no effect on AR negative, enz resistant PCa cells such as DU-145 and PC3? The authors should check these models, especially nearly half of the enz resistant CRPC patients are AR negative (Bluemn et al., 2017, PMCID: PMC5750052).

  1. In figure 1G, where is the ABT alone group? does ABT alone has similar effect as enz in enz-sensitive cell lines?

  1. In figure 2F, I am surprised to see that overexpression AR almost completely abolish the effect of ABT263. If ABT263 promotes AR degradation then it should still have partial effect even there is excessive amount of AR? If the results presented in figure 2F is correct, does this mean ABT263 will not be effective in PCa cell lines with much higher level of AR, such as VCaP and LNCaP/AR? the authors should also check these high AR models.

  1. The authors have showed that ABT could significantly decreased USP26 activity (Fig6H-J), then why knocking down USP26 itself reverse the effect of ABT on cell survival in Figure 5E? if both ABT and shUSP26 will lead to the down regulation of USP26 as shown in Figure 6, then why the combined treatment of ABT+shUSP26 is less effective then ABT alone or shUSP26 alone in Figure 5E? These parts is really confusing.

  1. For Figure 7B, it is very confusing that the DMSO treatment (black line) slowed down the tumor growth more than enz treatment (yellow line)? Or maybe the author mislabeled the figure?

Reviewer 2 Report

     Prostate cancer (PCa) is one of the most common non-cutaneous carcinoma of men in Western countries. Unfortunately, almost all patients ultimately die of castration-resistant prostate cancer (CRPC) during 3 years. Recently, the developed antiandrogen drug, Enzalutamide (Enz), has been as the standard treatment for CRPC patients. Therefore, it has become essential to identify molecular molecules and mechanisms of Enz-mediated chemoresistance in CRPC. Currently available findings indicated that targeting Enz-induced Bcl2 with inhibitor ABT263 could potential enhance Enz sensitivity in both Enz-sensitive and Enz-resistant PCa cells through induction of cellular ROS levels and suppression of USP26 activity, and then caused to the increase of ubiquitin/proteasome-dependent degradation of AR and ARv7 protein. The preclinical implication of the article indicated ABT263 combined with Enz may be a promising therapeutic option for PCa and resistant-Enz patients. Some aspects of the work, however, remain to be clarified.

  1. How many experiments were performed in Fig.1 H? Because the data was not presented with standard error in Fig. 1H.
  2. The authors suggested that AR and ARv7 were the critical molecules targeted by ABT263, however, oeAR expression was not significantly suppressed by ABT263 treated, as shown in Fig. 2F. The authors should classify and explain it.
  3. As shown in Fig. 2H, the authors described” We treated with two proteasome inhibitors, MG132 and Bortezomib, and found that both could partly block the decrease of AR and ARv7 expression in EnzS1-C4-2, EnzR1-C4-2, and 240 EnzR3-CWR22Rv1 cells”. Actually, there was no differences in AR and Arv7 expression by non-treatment compared with only treatment of MG132 and Bortezomib, respectively.
  4. The fluorescence image of ABT263 induced cellular ROS, as shown in Fig.3A was ambiguous, not clear and the fluorescence signal was not stronger than in control.
  5. The authors advised to provide the fluorescence image of ABT263 induced cellular ROS, negative control by NAC treated as shown in Fig. 3B and 3C.
  6. The protein intensity of internal control, GADPH was too various, not identical in Fig. 4A (left), 2A and 2H. These data seemed to reveal the bigger various ration of ABT263-mediated AR degradation in EnzS1-C4-2 cells. However, the data must be reproducible in every independent experiment.
  7. In manuscript, there are so many spelling mistakes in English words and bad grammar. Such as these mistakes: in abstract, “indentify”(line 20), “stainig”(line 23), “resisitant” (line 25, 31 and 192), in Methods, “determing”(line 95), in results, “Suplement” (line 194,197), “weather” (line 259) …so on. The authors should correct these mistakes and need an English Expert to assist the grammar revised.

Reviewer 3 Report

Since anti-apoptotic Bcl-2 appears to increase in progression to CRPC as well as resistance to Enz, the authors investigated the combination of ABT263 + Enz in several PCa cell lines sensitive and resistant to Enz. ABT263 alone appears to decrease AR and AR-V7 protein stability without decreasing AR mRNA.  There is the suggestion that 263 increases AR ubiquitination, thus increasing proteosomal degradation.  Data indicates 263 increases ROS production, which is counteracted by addition of antioxidant NAC. The suggestion is that increased ROS is important in AR decrease. The authors than identified DUBs that are known to regulate AR and their data suggests USP26 as a possible target for ROS. The idea is that 263 increase in ROS inhibits USP26 and lowers AR protein stability. ShRNA knockdown data supports a role for USP26. Addition of 263 with Enz appears to sensitize PCa cells (sensitive or resistant) to Enz. Xenograft data supports the 263 + Enz combination.

In general, the manuscript presents substantial, detailed, and high-quality data addressing a possible mechanism for 263 + Enz in PCa. There is support for the role of 263 increasing ROS, decreasing USP26 activity, and AR/AR-V7 protein. Perhaps a general schematic figure presenting the mechanism would be helpful.  Some questions and suggestions to improve this manuscript is presented below.

  1. I could not open the supplementary figures 1,2, 4, 6, and 7 (Microsoft office document image). Perhaps better if saved as JPEG image. Need to see these figures.
  2. Fig. 1: increase in Bcl-2 with EnzR does not occur in VCaP cells (1B). Please comment. In E-G, need to determine statistical significance between Enz, 263, and Enz+263. By eye, not a big difference. Find the CI values surprisingly low from data. Or presentation of data not clear.
  3. Fig. 2B: Exception is 22Rv1 where 263 appears to decrease AR/AR-V7 mRNA.
  4. For ip (Fig. 2L-M), would like to see neg control using non-specific Ig ip and Ub ib. Or could try Ub ip and AR ib. Does 263 increase total poly-Ub +MG? If 263 + Mg further increases poly-Ub, could be non-specific background. Same for ip result in Fig. 4.
  5. Fig. 6H, suggest a lighter exposure. Assume not adding HAUbVME the higher MW USP26 band not present. I, J: 263 decreases USP26 activity but NAC does not appear to antagonize. This goes against the proposed mechanism. Please comment.
  6. Fig. 7: Clarify treatment period (2 weeks every 2 days in methods; 1 week in fig. 7B). Difficult to see differences in 7E. If in 22Rv1 ARV7 stronger than ARfl by western, why so weak in IHC? Expectation is that 263 + Enz should see less AR/V7 but this is not what is observed.
  7. Discussion: Does increase in ROS inhibit other USP's listed? Perhaps it is a general inhibition of multiple DUBs that together lowers AR. Comment on clinical results if any using ABT263 in prostate cancer.
  8. Some word misspellings were noted.

Round 2

Reviewer 2 Report

The authors have provided some results and statements answer these questions in this manuscript. The spelling mistakes and English grammar are revised by a native English expert. So the manuscript can be accepted in this present form. 

Reviewer 3 Report

Authors have addressed my questions and revised version is acceptable. Only suggestion is to place supplemental figure 1 (Model) as figure 8 of main text so more readers can see.